# Reproductive Effects of *S. boulardii* on Sub-Chronic Acetamiprid and Imidacloprid Toxicity in Male Rats

**DOI:** 10.3390/toxics11020170

**Published:** 2023-02-12

**Authors:** Çiğdem Sevim, Erol Akpınar, Emrah Hicazi Aksu, Ali Doğan Ömür, Serkan Yıldırım, Mehtap Kara, İsmail Bolat, Aristides Tsatsakis, Robin Mesnage, Kirill S. Golokhvast, Sevgi Karabulut Uzunçakmak, Rabia Nilüfer Ersoylu

**Affiliations:** 1Department of Medical Pharmacology, Medicine Faculty, Kastamonu University, 37150 Kastamonu, Turkey; 2Department of Medical Pharmacology, Medicine Faculty, Ataturk University, 25240 Erzurum, Turkey; 3Department of Reproduction and Artificial Insemination, Veterinary Medicine Faculty, Kastamonu University, 37150 Kastamonu, Turkey; 4Department of Reproduction and Artificial Insemination, Veterinary Medicine Faculty, Atatürk University, 25240 Erzurum, Turkey; 5Department of Pathology, Veterinary Medicine Faculty, Atatürk University, 25240 Erzurum, Turkey; 6Department of Pharmeceutical Toxicology, Pharmacy Faculty, Istanbul University, 34452 Istanbul, Turkey; 7Department of Toxicology & Forensic Sciences, Medicine Faculty, University of Crete, 71003 Heraklion, Greece; 8Department of Medical and Molecular Genetics, King’s College London School of Medicine, Guy’s Hospital, London SE1 9RT, UK; 9Siberian Federal Scientific Center of Agrobiotechnology RAS, 630501 Krasnoobsk, Russia; 10Health Services Vocational School, Bayburt University, 69000 Bayburt, Turkey

**Keywords:** acetamiprid, imidacloprid, probiotic, *Saccharomyces boulardii*

## Abstract

The potential health-promoting effects of probiotics against intoxication by pesticides is a topic of increasing commercial interest with limited scientific evidence. In this study, we aimed to investigate the positive effects of probiotic *Saccharomyces boulardii* on the male reproductive system under low dose neonicotinoid pesticide exposure conditions. We observed that acetamiprid and imidacloprid caused a degeneration and necrosis of the spermatocytes in the tubular wall, a severe edema of the intertubular region and a hyperemia. This was concomittant to increased levels of 8-hydroxy-2′-deoxyguanosine reflecting oxidative stress, and an increase in caspase 3 expression, reflecting apoptosis. According to our results, *Saccharomyces boulardii* supplementation mitigates these toxic effects. Further in vivo and clinical studies are needed to clarify the molecular mechanisms of protection. Altogether, our study reinforces the burden of evidence from emerging studies linking the composition of the gut microbiome to the function of the reproductive system.

## 1. Introduction

Neonicotinoids are a pesticide group that has been increasingly popular in agricultural applications against pests in recent years. Imidacloprid, thiacloprid, clothianidin, thiamethoxam, acetamiprid, nitenpyram, and dinotefuran are the most used neonicotinoid class pesticides. Neonicotinoids exert their toxic effects via blocking the transmission of cholinergic signals and cause paralysis and hyperexitacion in insects. These chemicals affect the nAChR. The notoriety of neonicotinoids came to the fore with reports of their serious toxic effects on bees. Although it has recently been reported that neonicotinoids have very low toxic effects on mammals, studies have revealed that they can cause, at chronic exposures, serious toxic effects. These chemicals could accumulate in the environment [1,2]. Their toxic effects, particularly on honey bees and mammals, resulted in their gradual withdrawal from the market. Oxidative stress is one of the important cellular conditions that effects the male reproductive system, particularly spermatozoa [3,4]. It has been reported in different studies that neonicotinoids have negative effects on the male reproductive system, such as abnormal sperm morphology, sperm count decrease, and motility problems via different cellular pathways [3,5,6].

Occupational levels of various pesticides exposure may cause negative effects on the male reproductive system. All male reproductive system elements can be affected by pesticide exposure, such as the testes morphology, sperm motility and morphology disruptions, hormonal imbalance, disruption of seminal vesicles and alterations in prostate function. Oxidative stress induction, the inhibition of antioxidant activity and steroidogenesis may mainly cause these detrimental effects [7].

Two of the most commonly used neonicotinoid insecticides are acetamiprid and imidaprid. Neonicotinoids have also been reported to affect the male reproductive function [8,9,10]. It has been reported in different studies that neonicotinoids (particularly acetamiprid) affect reproductive system health via causing pathologic morphological changes in the ovary follicles, embryo quality reduction, and metabolic changes in ovary and genetic effects in female animals [11]. Imidacloprid could cause a decrease in semen quality, the disruption of hormonal changes and histopathological changes in the male reproductive system. It has been demonstrated that imidacloprid chronic exposure under NOAEL level doses causes detrimental effects on the male reproductive system in adults during the postnatal developmental stages [3]. In a study that exposed male rats to 12.5 mg/kg, 25 mg/kg and 35 mg/kg acetamiprid exposure for 90 days, it was reported that the levels of gonadotropin-releasing hormone (GnRH), follicle-stimulating hormone (FSH) and luteinizing hormone (LH) decreased dose-dependently. Additionally, the sperm numbers decreased and apoptosis occurred in the testes [5].

Chemicals that have an effect on the endocrine system and also adversely affect the reproductive system through this effect are known as endocrine disruptors. The most dramatic effect that neonicotinoid pesticides have on the reproductive system, which can be seen rapidly after agricultural sprays, was a serious decrease in egg laying efficiency in birds [12]. It has been suspected that different types of nAChRs receptors expressed on different organ cells, such as the nervous, prostate, and testes, make these organs the main target of acetamiprid and imidacloprid. These pesticides can also exert their toxic effects on other organs through oxidative stress [13,14] or metabolic disruptions, such as lipid accumulation mediated via the pregnane X receptor [15].

The gut microbiota play a key role in host protection against harmful pathogens and can protect from infections. In addition, it has important effects for host nutrition and immune system homeostasis [16]. The gut microbiota can also be a mediator in the toxic effects of pesticides because it possesses a large range of metabolic functions [17,18]. It is known that the use of probiotics, which dates back a long time, is good for health. While the use of probiotics is used to maintain the health status of healthy individuals, studies have also shown that probiotic support contributes to the recovery of various diseases, and could even protect against pesticide toxicity [19]. The use of fermented products containing probiotics accelerates the detoxification process of toxins in the body. *Bifidobacterium* is one of the first probiotics to be investigated and was discovered to have a disease-correcting effect. When used correctly, probiotics can have positive effects on almost all systems in the body, both in health and disease [20].

It is important for reproduction that the microbiota in the male reproductive system is balanced. In addition to the beneficial bacteria in the male reproductive system itself, the presence of bacteria that have a positive effect on different systems on the body, particularly the digestive system, is important for reproductive function. It was predicted that the intestinal microbiota may affect the male reproductive system through the intestinal-associated lymph nodes [21,22]. Microbial-associated molecular patterns (MAMPs) elements (lipopolysaccharide (LPS), lipoprotein acids, peptidoglycans, and lipoproteins) in the blood circulation could mediate this effect. These elements could activate the inflammation in the male reproductive system that can result in hypogonadism and decreased sperm quality. In addition, there are hypotheses that suggest the deterioration of the intestinal microbiota will cause pathologies in the male reproductive system through diabetes, triggering increased lipopolysaccharide-induced inflammation in the blood circulation, as well as hormonal system disorders; however, these studies are limited [21,22]. Many studies have been conducted on the role of probiotics in strengthening the detoxification mechanism, and as a result of these studies, the positive effects of probiotics have been demonstrated. Lactic acid bacteria (LABs), *Bifidobacterium* sp., and the yeast *Saccharomyces* (*S. cerevisiae var. boulardii*) are the first-line probiotics used for the aim of detoxification. In addition to their role in strengthening the antioxidant system, these probiotics also show their positive effects by preventing the passage of harmful substances. In addition to creating a physical barrier and accelerating the excretion of harmful substances by increasing intestinal motility, probiotics can accelerate the metabolism of harmful substances taken into the body through the enzymes they secrete [23].

To date, the toxicological evaluation of chemicals has focused on testing high-dose single chemicals in experimental animals to determine the safety margins and target organ(s) [24,25]. Unfortunately, this approach does not reflect the real-life exposure scenario in which individuals are exposed to multiple stressors at low doses at the same time [26]. In this study, we aimed to investigate the positive effects of probiotic *S. boulardii* on the male reproductive system under low dose neonicotinoid pesticides exposure conditions.

## 2. Materials and Methods

### 2.1. Animals

The study was carried out with 72 Wistar Albino male rats, weighing between 200–300 g, taken from the Experimental Research and Application Center (ATADEM) laboratory of Atatürk University. The water intake and pellet intake of the animals were ad libitum. Before starting the experiment, the rats were kept in cages for 1 week at normal room temperature (22 °C) in order to adapt to the environment. The ethics committee approval of the study was approved by the Animal Experiments Local Ethics Committee of Atatürk University with the document dated 11 January 2022 and numbered E-42190979-000-2200008858. A total of 72 male rats were randomized into 6 groups (5 experimental groups and 1 control group, 12 animals in each group). Before starting the experiment, all of the animals were weighed, and their weights were recorded. Imidacloprid (5.7 mg/kg), acetamiprid (12.4 mg/kg) and S. Boulardii (1 × 10^9^ CFU/day) were given orally to the animals, divided into groups, for 90 days [5,27,28].

The animals were sacrificed 90 days after injection. The rats were decapitated rapidly under deep anesthesia (Sevoflurane). One testis of each animal in each group was collected and fixed in 10% neutral formaldehyde for immunohistochemical analysis (Sigma-Aldrich, St. Louis, MO, USA). 

### 2.2. Collection of Samples

Following the decapitation procedure, the testes and cauda epididymidis of the rats were removed from the corpse and cleaned from the connective tissues, such as the adipose or connective tissues, with anatomical scissors and tweezers. Both the testes and cauda epididymidis were weighed and recorded as total testes weight (TTW) and total cauda epididymidis weight (TCEW).

### 2.3. Histopathological Examination

After the routine tissue follow-up, 4 µm thick sections were taken from the testicular tissue blocks for embedded in paraffin blocks and prepared on slides. The slides for the histopathological examination were stained with Hematoxylin-Eosin (H&E) and examined with a light microscope (Olympus BX 51, Tokyo, Japan). The sections were evaluated as absent (− 0 cells), mild (+ 1–5 cells), moderate (++ 6–10 cells), and severe (+++ 11 cells and above) according to the histopathological findings.

### 2.4. Immunohistochemical Examination

All of the sections were taken on adhesive (poly-L-Lysin) slides for the immunoperoxidase examination and a deparaffinized—Dehydrated procedure by passing through a xylol and alcohol series was performed. Then, the sections were washed in distilled water for 5 min. After that, they were washed with phosphate buffer solution (PBS, pH 7.2) for 5 min, kept in 3% H_2_O_2_ for 10 min, and endogenous peroxidase inactivated. Then, the tissues were boiled in 1% antigen retrieval (citrate buffer (pH +6.1) 100×) solution and allowed to cool at room temperature. After washing for 5–10 min in PBS, the sections were incubated for 5 min with a Protein block that is compatible with all primary and secondary antibodies to prevent nonspecific background staining. After the excess of the block solution remaining on the tissue sections at the end of the incubation was poured, the primary antibodies (8-OHdG Catalog Number:sc-66036 Diluent Ratio: 1/100, US and Caspase-3 Catalog Number:sc-56053 Diluent Ratio: 1/100, US) and PBS in the control group were dripped without washing. In accordance with the primary antibody, it was incubated at room temperature for 1 h or at +4 °C for 1 night. After incubation, the sections were washed with PBS again, 2 times for 5 min, and incubated with a biotinized secondary antibody for 10–30 min at room temperature. The sections washed again with PBS and streptavidin-peroxidase for 10–30 min. Then, 3-3′ Diaminobenzidine (DAB) chromogen was dripped onto the sections for 8-OHdG and Aminoethyl carbazole (AEC) chromogen was dripped onto the sections for Caspase-3 for 5–10 min, depending on the chromogen removal. Between 1–2 min of Mayer’s hematoxylin was performed for background staining. Then, they were washed with tap water. Alcohol and xylol series were conducted, covered with a coverslip and examined with a light microscope (Zeıss AXIO, Zeiss, Germany) [29,30,31,32].

### 2.5. Collection of Samples, Sperm Evaluation and Epididymal Sperm Parameters

One of the cauda epididymidis was used to obtain the semen sample for each animal. For this purpose, a randomly selected cauda epididymidis was minced in a Petri dish, including 5 mL of physiological saline. To provide the migrations of the spermatozoa from the cauda epididymidis to the fluid, a 5 min incubation period was conducted at the warmed stage (at 35 °C). Following the incubation period, the cauda epididymidis residue was removed by using anatomical tweezers from the Petri dish. The fluid remaining in the Petri dish was used as the semen sample. The evaluation of the semen was conducted using routine spermatological parameters, including motility, density of sperm cells, dead sperm rate and a morphological examination of spermatozoa. To evaluate the percentage of sperm motility, with a Zeiss Primo Star (Carl Zeiss Inc, Oberkochen, Germany) light microscope equipped with the heated stage was used. Briefly, a slide was placed on a heated stage warmed up to 35 °C and placed on a conventional light microscope. Approximately 20 µL of semen sample was dropped on the slide. The percentage of sperm motility was detected through a visual investigation of the sample. To estimate the sperm motility, three randomly selected different fields from each sample were evaluated. The average of the three field estimations was calculated as the final motility score of the sample [33,34]. To determine the sperm cells’ concentration, the method described in our previous study [35] was used. Briefly, the semen sample was diluted as the rate of 1/100 with eosin solution (2 g dry eosin dye and 3 g sodium citrate solved in 100 mL of distilled water) in an Eppendorf tube. The Eppendorf tubes were vortexed at 2500 rpm for 15 s and the sperm suspension was transferred into the counting chambers of a Thoma chamber. Then, the sperm cells in both chambers were counted under the conventional light microscope (Zeiss Primo Star) at the magnification of 4009. To determine the percentage of the morphological abnormalities of the spermatozoa, the method (with a little modification by using only eosin dye instead of eosin-nigrosin dye) described by Turk et al. was used [36]. Briefly, the slides were stained with eosin dye. Then, the slides were evaluated under a light microscope at 4009 magnification with the help of immersion oil (immersion oil for microscopy type A, no: 1.515; Nikon, Tokyo, Japan). Two hundred and fifty spermatozoa from each slide were observed and the percentages of the sperm head, sperm mid, sperm tail and total abnormality of spermatozoa were stated.

### 2.6. Statistic Analyses

The spermatological data in the study were expressed as mean value ± SEM. The spermatological parameters were analyzed using a one-way analysis of variance test. In addition, a post-hoc Tukey test was used. The SPSS 13.0 program was used for the statistical analysis in the histopathological examinations and the data were evaluated with *p* < 0.05 were considered significant. A Duncan test was used for the comparison between groups. The non-parametric Kruskal-Wallis test was used to detect the group interaction, and the Mann Whitney U test was used to determine the differences between the groups. In order to determine the intensity of the positive staining from the pictures obtained as a result of immunohistochemical staining, 5 random areas were selected from each image and evaluated in the ZEISS Zen Imaging Software program. The data were statistically defined as mean and standard deviation (mean ± SD) for % area. The one-way ANOVA followed by the Tukey’s test were performed to compare the positive immunoreactive cells and immunopositive stained areas with the healthy controls. As a result of the test, a value of *p* < 0.05 was considered significant and the data were presented as mean ± SD.

## 3. Results

### 3.1. Pathologic Analyses

#### 3.1.1. Histopathological Findings

The testicular tissues show severe degeneration and necrosis in the spermatocytes at the tubular wall; severe edema in the tubular region and hyperemia in the blood vessels were observed in the ACE and IMI groups, while slight degeneration and necrosis were observed in the spermatocytes at the tubular wall and hyperemia in the blood vessels in the IMI + PRO and ACE + PRO group. The exposed animal groups testes tissues histopathological findings are listed in Table 1 (Figure 1). The scoring of the histopathological findings are expressed in Table 2.

#### 3.1.2. Immunohystochemical Findings

It was observed that 8-OHdG and Caspase expression were detected to be severely positive in the IMI and ACE groups, and slightly positive in the IMI + PRO and ACE + PRO groups. The immunohistochemically analysis results of 8-OHdG and Caspase 3 are explained in Table 3 (Figure 1). The scoring results are shown in Table 4.

### 3.2. Reproductive Organ Weights and Epididymal Sperm Parameters

Sperm count and morphology evaluations were performed according to the World Health Organization (WHO) guideline [34]. The total testis weight (TTW) and total epididymis weights (TCEW) are shown in Table 5. No significant difference (*p* > 0.05) was found between the control and the other groups. The sperm motility percentages are presented in Table 5. In terms of the sperm motility rate, both the IMI + PRO and ACE + PRO group had lower values than those in the control group. However, no significant difference (*p* > 0.05) was found between the control and the other groups. The sperm cell densities for all of the groups studied are presented in Table 5. It was significantly (*p* < 0.05) higher in the controls than the other groups. Epididymal sperm abnormalities were classified as head, mid-piece, tail and total sperm abnormality. There was no significant difference (*p* > 0.05) among all the groups in terms of all of the abnormalities concerned. The values of all the sperm abnormalities of the groups are presented in Table 6. The dead sperm rates are presented in Table 6. There was no difference (*p* > 0.05) between the control and the other groups.

## 4. Discussion

The role of probiotics in protecting against the effects of environmental pollutants is still being studied and is not well understood. Some studies have suggested that certain strains of probiotics may have protective effects against the toxicity of certain pollutants, such as heavy metals. However, the evidence is limited, and more research is needed to establish a clear link between probiotics and the protection against environmental pollutants. In this study, we found that two common neonicotinoid pesticides, acetamiprid and imidacloprid, induce the oxidative stress-related cell death of the sperm and testes, as well as vessel morphological disruptions. We observed that acetamiprid and imidacloprid cause severe degeneration and necrosis in spermatocytes. However, with probiotics, these findings were decreased. Additionally, severe edema was detected at the intertubular region and severe hyperemia was detected in the blood vessels with neonicotinoids exposure. While we found severe intracytoplasmic 8-OHdG and Caspase-3 expressions in acetamiprid and imidacloprid groups, adding probiotic reduced these higher expressions and detected slight intracytoplasmic 8-OHdG and Caspase-3 expressions. According to these results, the probiotic treatment groups are significantly different compared to the acetamiprid and Imidacloprid treated groups (*p* < 0.05).

Apoptosis is a programmed cell death process and, in many cases, is necessary for the organism to maintain homeostasis. It is also known that at various stages of mammalian spermatogenesis, germ cell apoptosis occurs to eliminate abnormal spermatogenic cells and subsequently maintain the normal quantity and quality of sperm. There are many factors that affect apoptosis in germ cells. Toxic substances are also among the factors responsible for this induction. The excessive apoptosis of these cells leads to vital conditions, such as oligozoospermia and azoospermia. In many organisms, well-conserved caspases are processed into active forms in cells undergoing apoptosis and thus play a crucial role in the transmission of apoptotic signals in cells that will die, and its consecutive activation is considered as one of important keys in the cellular apoptosis process. Their activation is due to self-proteolysis and/or the effects of other proteins. The significant increase in the caspase 3 density in the acetamiprid and imidacloprid groups suggests that testicular cells are driven to apoptosis at a high rate. It has drawn our attention that probiotics, which reduce this intense increase, can reverse the apoptosis process to a certain extent [37,38,39].

Hartman et al. reported that one of the neonicotinoids disrupts the male reproductive system with gestational exposure. Neonicotinoid exposure causes the sperm count decrease and male reproductive organ morphological alterations, ATP depletion, and a meiotic defect related with chromatin disorganization [40]. In another study, chronic 8 mg/kg imidacloprid treatment caused severe disruptions in the male rat reproductive system through abnormal sperm morphology, sperm motility abnormality and decreased levels of testosterone (T) and Glutathione (GSH). Increased apoptosis of germ cells and seminal DNA fragmentation were also observed. Antioxidant molecule depletion could be a sign of oxidative stress-related apoptosis [6]. Our finding in the imidacloprid and acetamiprid groups are consistent with the findings of these studies.

Gu et al. exposed mouse sperms directly to imidacloprid and acetamiprid. They observed a decreased fertilization capacity without any chance of sperm motility. The sperm motility and DNA integrity were not significantly affected by a high exposure dosage (5 mM for 30 min) of acetamiprid and imidacloprid [8]. In our study, we did not find significant different abnormal sperm counts or morphologies between the groups via microscopic evaluation.

In a 56 days oral imidacloprid exposure study, imidacloprid caused decreased animal and male reproductive organ weights, a decreased testosterone level and induced apoptosis of the testis tissue. These findings were related to the increase in the cellular oxidative stress parameter malondialdehyde (MDA), as well as the decrease in the Glutathione (GSH) and total antioxidant capacity (TAC) [41]. In accordance with this study’s results, in our study, the male rat reproductive organ and sperm apoptosis was associated with an oxidative stress increase. Oxidative stress caused by neonicotinoids can cause damage to cellular macromolecules, such as DNA, lipids, and proteins, and then cell death may occur via apoptotic or necrotic mechanisms [42]. Studies have shown that degeneration occurs in the seminiferous tubules with pesticide exposure and that the expression of antioxidant enzymes, such as Glutathione peroxidase, is suppressed [43]. 8-hydroxy-2-deoxyguanosine is recognized as a biomarker for the oxidative damage of DNA and its levels are known to increase due to exposure to physical, chemical or biological agents. Oxidative stress-related DNA damage occurred in our study with an increase in the 8-OHdG levels. This was the case in the groups exposed to acetamiprid and imidacloprid and we thus made the hypothesis that there was oxidative damage in the testicular cell DNA. The relatively low degree of this damage in the groups supplemented with probiotics gave hope that probiotics could reduce oxidative damage.

In a chronic study, in which lizards were exposed to imidacloprid at 0, 10, 50, 100 mg/kg bw doses and dose-dependent apoptosis induction, spermatogenesis disruption and a decrease in the sex hormone levels were found [44]. In another study, acetamiprid exposure for 45 days caused a decrease in the body weight gain, a decrease in the testes, epididymis, and seminal vesicles organ gains and a spermatids count decrease; as well as a sperm count decline, a significant decrease in the sperm motility and testosterone levels compared to control group were found. Additionally, significant increases in the abnormal and dead sperm and TBARS levels were found [13].

Terayama et al. demonstrated that 180 days of chronic administration of acetamiprid resulted in severe testis tissue degeneration. An interesting finding of this study is that the adult testes effected more than the immature mice. This could be related to an acetamiprid accumulation in the tissues [45]. According to the results of these studies, acetamiprid and imidacloprid exert serious toxic effects on the male reproductive system.

The stability of the microbiota is important for the reproductive system in living beings. In particular, studies have shown that probiotic supplementation has positive effects on the female and male reproductive systems, and it has even been reported that it can be considered as a therapeutic option [22,46]. It has been demonstrated that different probiotic supplementations protect the male reproductive system from different stressors. Stress can induce oxidative stress-related cell death in male germ cells and reduces the fertility potential. When applied in appropriate amounts and at appropriate times, probiotics provide great support in protecting organs against stress factors [47]. There have been limited studies on the positive effects of probiotics against neonicotinoid pesticides’ toxic effects. All of these studies include different organs and systems, such as the liver, kidney and immune system; however, there have been no data concerning probiotics’ effects on the reproductive system against neonicotinoids toxicity [28,48,49,50]. This is the first study to evaluate and report the positive effects of probiotic S.boulardii supplementation against imidacloprid and acetamiprid toxicity in male rat reproductive organs. According to our results, S.boulardii supplementation ameliorates the toxic effects of acetamiprid and imidacloprid as degeneration and necrosis has been detected/observed in the spermatocytes at the tubular wall, severe edema was detected at the intertubular region, hyperemia was detected in the blood vessels, as well as 8-OHdG-associated oxidative stress in accordance with apoptosis.

Not being able to observe the effects on the sperm after 90 days in our study is one of the limitations of this study. It is obvious that there is a significant degeneration in the testicular tissue at these doses, but one reason why this was not reflected in the sperm count, motility and density, which will be a clinical indicator, may be the insufficient time in which the analyses were performed. In the cauda epididymis, sperm can survive for up to 60 days under experimental conditions. Therefore, the degenerations in the testis will have escaped this effect, for example, in sperm that were produced 1–2 weeks before the degeneration process. A longer study is needed to more clearly see the clinical repercussions of exposures at low doses. This was a limiting factor in our study.

## 5. Conclusions

In conclusion, the induction of oxidative stress-related apoptosis is a mechanism of male reproductive system disruption via neonicotinoids, and for protection against these effects, probiotic use emerges as an important option. To clarify the detailed molecular mechanisms of protection further in vivo, clinical studies are needed because of the numerous gaps in the knowledge concerning the applicability of studies showing the effects of probiotics in animal models to humans. This includes species-specific differences (the effects of probiotics can vary between species), dosage and formulation (dosages used in animal studies may not be equivalent to dosages used in human trials), gut microbiome diversity (the gut microbiome in humans is much more complex and diverse than in animal models), and study design as animal studies may not accurately reflect the real-world conditions in which humans use probiotics, such as dietary habits and lifestyle factors. Altogether, our study provides evidence for the potential use of probiotics as a strategy to mitigate the negative impact of pesticides on reproductive health.

## Figures and Tables

**Figure 1 toxics-11-00170-f001:**
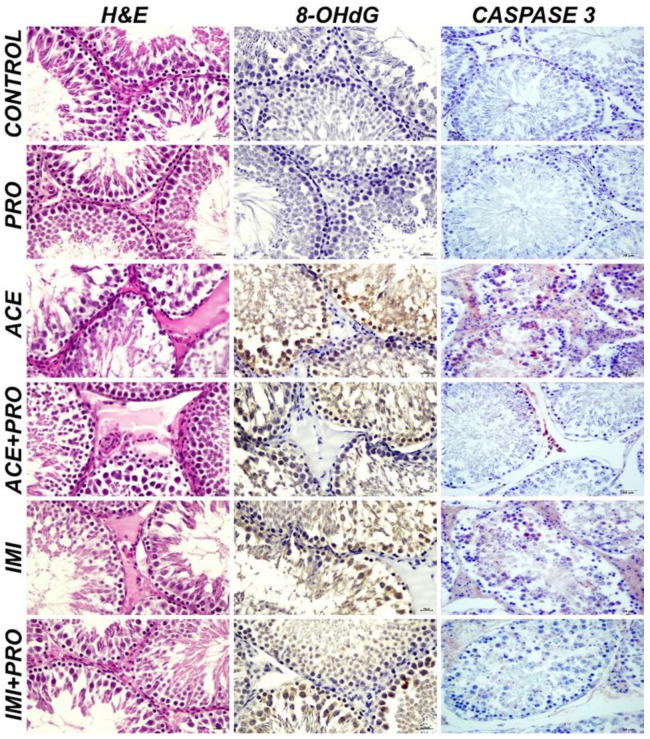
Testicular tissue, histopathological findings, H&E; 8-OHdG expressions (DAB chromogen), Caspase 3 expressions (AEC chromogen), IHC-P, Bar: 20 µm. (PRO: Probiotics, ACE: Acetamiprid, ACE + PRO: Acetamiprid + Probiotics, IMI: Imidacloprid, IMI + PRO: Imidacloprid + Probiotics, H&E: Hematoxylin and eosin, DAB: 3,3-Diaminobenzidine, AEC: 3-Amino-9-Ethylcarbazole, 8-OHdG: 8-Hydroxy-2′-deoxyguanosine, IHC-P: Immunohistochemistry).

**Table 1 toxics-11-00170-t001:** Experimental groups histopathological findings (H&E: Hematoxylin and eosin, Bar: 20 µm).

Animal Groups	Findings
Control	Normal morphology data
Probiotics	Normal morphology compared to control group
Acetamiprid	Severe degeneration and necrosis has been detected/observed in spermatocytes at tubular wall; severe edema detected at intertubular region and hyperemia detected in blood vessels.
Acetamiprid + Probiotics	Slight degeneration and necrosis has been detected/observed in spermatocytes at tubular wall; edema detected at intertubular region and slight hyperemia detected in blood vessels. These results are significantly different compared to Acetamiprid group (*p* < 0.05).
Imidacloprid	Severe degeneration and necrosis has been detected/observed in spermatocytes at tubular wall and related with necrosis thinning of tubular wall detected; severe edema detected at intertubular region and severe hyperemia detected in blood vessels.
Imidacloprid + Probiotics	Slight hyperemia was detected in tubular wall and at vessels. These results are significantly different compared to Imidacloprid group (*p* < 0.05).

Significant differences at *p* < 0.05 compared to control group.

**Table 2 toxics-11-00170-t002:** Scoring of histopathological findings (Sections were evaluated as − (absent, 0 cells), + (mild, 1–5 cells), ++ (moderate, 6–10 cells), and +++ (severe, 11 cells and above)).

	Control	Probiotics	Acetamiprid	Acetamiprid + Probiotics	Imidacloprid	Imidacloprid + Probiotics
Degeneration at spermatocytes	−	−	+++	+	+++	+
Spermatocyte necrosis	−	−	+++	+	+++	−
Hyperemia in vessels	−	−	+++	+	+++	+
Tubular wall thinning	−	−	+++	+	+++	+
Edema at intertubular gap	−	−	+++	+	+++	−

**Table 3 toxics-11-00170-t003:** Experimental groups immunohistochemically findings of 8-OHdG expressions (DAB chromogen) and Caspase 3 expressions (AEC chromogen) (DAB: 3,3-Diaminobenzidine, AEC: 3-Amino-9-Ethylcarbazole, 8-OHdG: 8-Hydroxy-2-deoxyguanosine, IHC-P: Immunohistochemistry).

Animal Groups	Findings
Control	When testicular tissues are examined immunohistochemically; Expressions of 8-OHdG and Caspase-3 were evaluated as negative
Probiotics	Expressions of 8-OHdG and Caspase-3 were evaluated as negative
Acetamiprid	Severe intracytoplasmic 8-OHdG and Caspase-3 expressions were detected.
Acetamiprid + Probiotics	Intermediate intracytoplasmic 8-OHdG and Caspase-3 expressions were detected. These results are significantly different compared to Acetamiprid group (*p* < 0.05).
Imidacloprid	Severe intracytoplasmic 8-OHdG and Caspase-3 expressions were detected.
Imidacloprid + Probiotics	Slight intracytoplasmic 8-OHdG and Caspase-3 expressions were detected. These results are significantly different compared to Imidacloprid group (*p* < 0.05).

Significant differences at *p* < 0.05 compared to control group.

**Table 4 toxics-11-00170-t004:** Immunohistochemistry scoring results of 8-OHdG and Caspase-3 [mean ± standard error of means (S.E.M.)].

Animal Groups	8-OHdG	Caspase 3
Control	19.82 ± 0.18	20.33 ± 0.7
Probiotics	19.13 ± 0.56	19.76 ± 0.53
Acetamiprid	69.57 ± 1.18 *	60.38 ± 0.92 *
Acetamiprid + Probiotics	42.95 ± 1.09 *	37.55 ± 0.81 *
Imidacloprid	58.12 ± 1.26 *	50.69 ± 1.23 *
Imidacloprid + Probiotics	30.06 ± 0.85 *	25.48 ± 0.79 *

*: Significant differences at *p* < 0.05 compared to control group.

**Table 5 toxics-11-00170-t005:** Total testes, total cauda epididymis weights and epididymal sperm parameters [mean ± standard error of means (S.E.M.)].

Groups	Total Testis Weight (TTW) (g)	Total Cauda Epididymis Weights (TCEW) (g)	Motility (%)	Density (10^3^/mL)
Control	3.68 ± 0.14	0.56 ± 0.01	58.12 ± 2.66	60,000 ± 2500
Probiotics	3.86 ± 0.16	0.59 ± 0.03	58.75 ± 3.62	42,500 ± 1336.30
Acetamiprid	3.85 ± 0.15	0.57 ± 0.01	62.50 ± 2.83	50,625 ± 1132.90
Acetamiprid + Probiotic	3.93 ± 0.15	0.59 ± 0.01	53.75 ± 2.63	43,125 ± 1619.49
Imidacloprid	3.65 ± 0.11	0.57 ± 0.01	62.12 ± 2.28	59,687 ± 2929.12
Imidacloprid + Probiotic	3.73 ± 0.08	0.58 ± 0.02	51.25 ± 3.09	49,375 ± 2902.81

**Table 6 toxics-11-00170-t006:** Number of abnormal sperm count [mean ± standard error of means (S.E.M.)].

Groups	Dead Sperm (%)	Head Abnormality (%)	Mid-Piece Abnormality (%)	Tail Abnormality (%)
Control	14.57 ± 2.36	2.07 ± 0.46	0.14 ± 0.09	0.42 ± 0.13
Probiotics	20.00 ± 1.77	2.50 ± 0.72	0.62 ± 0.20	1.00 ± 0.40
Acetamiprid	14.64 ± 1.01	2.07 ± 0.57	0.14 ± 0.09	0.57 ± 0.25
Acetamiprid + Probiotic	16.18 ± 0.85	2.56 ± 0.30	0.37 ± 0.15	0.93 ± 0.34
Imidacloprid	14.75 ± 0.79	1.12 ± 0.22	0.18 ± 0.13	0.62 ± 0.22
Imidacloprid + Probiotic	14.43 ± 1.43	1.75 ± 0.26	0.18 ± 0.09	0.31 ± 0.18

## Data Availability

The data presented in this study are available on request from the corresponding author.

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
