# Peer review of "Reproductive Effects of S. boulardii on Sub-Chronic Acetamiprid and Imidacloprid Toxicity in Male Rats"

_toxics, 2023, doi:10.3390/toxics11020170_

Round 1
Reviewer 1 Report
Dear Authors,
I am sending my opinion attached.
Best regards

Author Response
Thank you very much for the comments of the esteemed reviewer, they have been a light for us to improve our article.
You can find the attached file for responses

Reviewer 2 Report
Please see attached document

Author Response

(The authors gave the same response as above.)

Reviewer 3 Report
The manuscript was aimed to investigate the positive effects of probiotic Sacharomyces Boulardii on male reproductive system under low dose neonicotinoid pesticides exposure conditions. The authors observed that acetamiprid and imidacloprid caused a degeneration and necrosis of spermatocytes in the tubular wall, a severe edema of the intertubular region and a hyperemia. There were observed the increase of levels of 8-hydroxy-2’-deoxyguanosine
reflecting oxidative stress, and an increase in in caspase 3 expression reflecting apoptosis. The authors stated that S. Boulardii supplementation mitigates the toxics effects of neonicotinoids. The further studies are needed to clarify molecular mechanisms of protection.
The paper is interesting and prepared correctly. The results obtained are new and add more information to the science. They touch the current problem of the decreased fertility of men after exposure to endocrine disruptors. This problem is very important from the point of view of public and reproductive health. The manuscript is recommended for publication after minor revision. Following are specific comments:
The order of groups in all Tables should be the same, in this way is easier to read.
In Discussion the authors should discuss their own results in the comparison with other papers instead of describes each paper of other authors only.
In Table 6 results of sperm abnormality after probiotic supplementation are the worst among all control and experimental groups, especially in the percent mid-piece and tail abnormality. Such results might be statistically significant, but there are no finding about this. The statistical analysis is necessary in such study. Please, provide the results of statistical analysis.
Moreover, there are several printing and grammar errors, which require corrections.
Author Response

(The authors gave the same response as above.)

Reviewer 4 Report
This study is relatively novel in that it attempted to show that neonicotinoids, which were previously thought to be less toxic to mammals including humans, cause spermatogenesis defects in rats and that probiotics are effective in reducing their toxicity. However, it is uncomfortable to note that while ACE- and IMI-induced spermatogenesis defects have been observed in testicular histology, no abnormalities in testis or epididymal weight or epididymal spermatozoa have been observed. And it is very strange that the authors do not consider at all that the administration of neonicotinoids had no effect on sperm productivity.
Furthermore, the authors' analysis of testicular tissue sections, the results of which they presented in order to argue their conclusions, is not sufficient to describe the experimental method, and it cannot be judged whether the method was sufficient to argue the results. It is unclear how the results should be read without clarification on at least the following points.
I don't understand what the "Some of …, and some …” in lines 155-157 refers to. 72 rats were divided into 6 groups of 12 rats per group, so how were they used in the analysis? The authors need to describe exactly what they mean.
Furthermore, in the next paragraph (4.1), lines 163-165, "For the pathological examinations, one of the testes was kept in Bouin’s solution while another one was kept in deep freeze immediately (at 20 °C) for biochemical assays by classifying for their groups." but I was confused as to whether the testes were fixed in 10% neutral formalin in the previous paragraph. It says that one testis was cryopreserved for biochemical measurements to classify groups, but it is unclear what was measured for this biochemical measurement and I cannot find the results.
In addition, although histological studies are being conducted, a semi-quantitative method called Jonsen's Score Count exists for the evaluation of spermatogenesis by testicular sections, so an accepted method should be adopted rather than an original method.
In paragraph 4.3, there is information on immunostaining methods, but the information on 8-OHdG and Caspase-3 antibodies is missing. 8-OHdG antibodies require caution because H2O2 treatment may affect them, and Caspase-3 is activated when its precursor protein is digested. Since Caspase-3 is activated by digestion of precursor proteins, it is necessary to explain which stage of Caspase-3 labeling the antibodies used. In some cases, activation cannot be determined by staining tissue sections and must be examined by Western blotting.
In the presentation of the results, the contents of Table 1 and 3 should be included in the text; Table 4 contains the results of quantification, but we could not find a method for this quantification.
Author Response

(The authors gave the same response as above.)

Reviewer 5 Report
This study reinforced the burden of evidence from emerging studies linking the composition of the gut microbiome to the function of the reproductive system. This study was well designed. Data are sound. A revision is suggested.
1. Please emphasize the role of oxidative stress in this model.
2. Please discuss the limitations and the clinical implications of this study.
3. How many animals were used in each assay. Please mention in fig led.
4. Please discuss the role of apoptosis in this model.
Author Response

(The authors gave the same response as above.)

Round 2
Reviewer 2 Report
The Authors have addressed properly all the comments indicated, then in my opinion the manuscript can be accepted.
Reviewer 4 Report
I am not completely satisfied with the authors' response, but I agree with publication because I think the manuscript has been improved and is consistent with the journal's objectives.
Reviewer 5 Report
My questions had been well addressed . This submission is acceptable in this vision.